# Firing Alterations of Neurons in Alzheimer’s Disease: Are They Merely a Consequence of Pathogenesis or a Pivotal Component of Disease Progression?

**DOI:** 10.3390/cells13050434

**Published:** 2024-02-29

**Authors:** Nikolaos P. Tzavellas, Konstantinos I. Tsamis, Andreas P. Katsenos, Athena S. Davri, Yannis V. Simos, Ilias P. Nikas, Stefanos Bellos, Panagiotis Lekkas, Foivos S. Kanellos, Spyridon Konitsiotis, Charalampos Labrakakis, Patra Vezyraki, Dimitrios Peschos

**Affiliations:** 1Department of Physiology, Faculty of Medicine, School of Health Sciences, University of Ioannina, 451 10 Ioannina, Greece; 2Department of Neurology, Faculty of Medicine, School of Health Sciences, University Hospital of Ioannina, 455 00 Ioannina, Greece; 3Medical School, University of Cyprus, 2029 Nicosia, Cyprus; 4Department of Biological Applications and Technology, University of Ioannina, 451 10 Ioannina, Greece

**Keywords:** Alzheimer’s disease, neurophysiology, pyramidal neurons, interneurons, LTP, dendritic spine, neuronal firing properties

## Abstract

Alzheimer’s disease (AD) is the most prevalent neurodegenerative disorder, yet its underlying causes remain elusive. The conventional perspective on disease pathogenesis attributes alterations in neuronal excitability to molecular changes resulting in synaptic dysfunction. Early hyperexcitability is succeeded by a progressive cessation of electrical activity in neurons, with amyloid beta (Aβ) oligomers and tau protein hyperphosphorylation identified as the initial events leading to hyperactivity. In addition to these key proteins, voltage-gated sodium and potassium channels play a decisive role in the altered electrical properties of neurons in AD. Impaired synaptic function and reduced neuronal plasticity contribute to a vicious cycle, resulting in a reduction in the number of synapses and synaptic proteins, impacting their transportation inside the neuron. An understanding of these neurophysiological alterations, combined with abnormalities in the morphology of brain cells, emerges as a crucial avenue for new treatment investigations. This review aims to delve into the detailed exploration of electrical neuronal alterations observed in different AD models affecting single neurons and neuronal networks.

## 1. Introduction

Alzheimer’s disease (AD) is a devastating neurodegenerative disorder, yet its precise origins and pathogenic mechanisms remain unclear. Extensive global research endeavors have yielded numerous insights regarding the genetic and molecular events underlying the accumulation of amyloid beta (Aβ) oligomers and the disruption of neuronal cytoskeleton due to Tau protein dysregulation. These processes have been demonstrated to induce synaptic dysfunction and abnormal neuronal excitability, forming the basis for AD’s clinical manifestations [1]. However, the exploration of alterations in the neurophysiological properties of individual neurons and neuronal networks, traditionally viewed as secondary, has received limited attention within the context of the disease. These understudied facets may provide crucial insights that could contribute to a more comprehensive understanding of AD pathogenesis [2].

The alteration of electroencephalography (EEG) patterns in patients with AD is well established, primarily characterized by an elevation in theta (θ) and delta (δ) rhythms within frontal cortical regions and a decline in alpha (α) rhythms in parietal and occipital lobes, accompanied by disrupted synchronization [3]. Emerging experimental evidence also highlights the significance of gamma (γ) and θ oscillations in learning and memory within the hippocampal formation, particularly their interplay [2,4]. This suggests the potential for manipulating brain activity to enhance memory formation. Moreover, recent decade-spanning discoveries have highlighted the pivotal role of neuronal and brain activity as potential modulators of AD pathology. Sensory and electrical brain stimulation have demonstrated the capacity to diminish Aβ accumulation, alter microglia behavior, and reinstate γ oscillations in AD models, culminating in cognitive function improvement [5,6,7].

Taken together, these studies underscore the significant role of brain activity in the pathogenesis of AD and suggest the therapeutic potential of strategies focused on precise modulation of neuronal activity. Nevertheless, the present understanding of the underpinning mechanisms of AD-related alterations in neuronal and network activity remains incomplete, posing challenges in predicting the outcomes of potential interventions. This review aims to underline the neurophysiological alterations of the neurons and circuits associated with AD pathology, with the goal of laying the groundwork for a deeper understanding of activity changes. This knowledge can potentially be harnessed in the future to enhance the efficacy of non-invasive brain stimulation treatments.

We performed an online search to retrieve studies that specifically examined the electrical activity of cells in the context of AD. From the extensive pool of research that centered on long-term potentiation or depression (LTP or LTD) and EEG, we selectively included studies that contributed meaningful insights into the microscale electrophysiology of neurons for this review.

## 2. Neuronal Excitability in AD

### 2.1. Pyramidal Neurons

A series of comprehensive studies have yielded valuable insights into the complexity of pyramidal neuron activity, providing a deeper understanding of diverse facets of neural processes on AD. Pyramidal neurons in the hippocampus possess long membrane time constants (τ) and exhibit high input resistance along with high specific resistivity on their soma and dendrites. Additionally, they demonstrate a low threshold for initiating spikes in the soma-axon hillock region. These cells are differentially regulated by the synergistic action of glutamate and acetylcholine (ACh), acting on metabotropic receptors. This suggests that the hippocampus functions through parallel pathways that handle different types of information [8,9,10,11]. In terms of electrophysiology, two types of principal cells within the hippocampus are distinguished based on their electrophysiological responses. Regular-spiking neurons manifest single spikes early in a train of stimuli, transitioning to bursts later, while bursting neurons initiate bursts early in the train followed by single spikes (Figure 1).

Regular-spiking pyramidal neurons exhibit higher input resistance and a greater threshold for depolarizing the cell membrane compared to their bursting counterparts. However, they display a smaller post-spike afterdepolarization [12]. Early studies have shown that when exposed to ACh, pyramidal cells displayed distinct behavior during spiking and bursting. Low ACh levels led to robust dendritic depolarization and calcium influx during bursting, while higher ACh concentrations resulted in weaker, multiple dendritic depolarizations during spiking, with minimal calcium influx. The difference in membrane depolarization played a significant role, with spiking causing short, small dendritic depolarizations, and bursting leading to a prolonged dendritic depolarization. Since calcium is crucial for synaptic plasticity, these findings suggest that spiking may not be ideal for processes like memory storage and recall, whereas bursting could be more effective. This transition between bursting and spiking may have implications for memory-related biological networks [13].

Neuronal hyperexcitability and epileptiform activity in the course of AD were noted decades ago, but the exact underlying mechanism remained unclear for many years [14,15]. Experiments in transgenic mouse models of AD have demonstrated a connection between high levels of Aβ and epileptiform activity as well as cognitive deficits. These findings suggest that Aβ peptides might play a role in the cognitive decline observed in AD, potentially by inducing aberrant neuronal activity [14]. Early hippocampal hyperactivity in AD mice highlights the crucial role of soluble Aβ in this dysfunction. Notably, in young mice, before plaque formation, hyperactive neurons gradually increase, possibly due to soluble Aβ. In hippocampal neurons of the CA1 area, it is suggested that soluble Aβ and amyloid plaques alter the function of neuronal circuits by disrupting the balance of synaptic excitation and inhibition [14]. Hyperactive neurons are mainly located adjacent to Aβ plaques whereas silent neurons away from them [16,17]. A γ-secretase inhibitor LY-411575 reduces soluble Aβ levels and restores normal neuronal function when administered acutely [18]. The Tau protein also affects neuronal excitability, since it has been shown that its accumulation in CA1 pyramidal neurons causes an increased excitability at more hyperpolarized membrane potentials, while excitability decreases at more depolarized potentials. This change toward greater sensitivity to hyperpolarized potentials in the voltage response of K^+^ currents appear to be linked to the phosphorylation of potassium and sodium channels (K_v_7 and Na_v_1.6 types). These findings underscore the role of phosphorylation in dynamically modulating neuronal excitability across a range of membrane potentials.

Voltage-gated sodium and potassium channels are critical components of the neuronal signaling system in nervous system. These specialized ion channels are central to shaping neuronal electrical properties, controlling excitability, and facilitating signal transmission. Sodium channels initiate and propagate action potentials by responding to depolarization, while potassium channels regulate membrane potential, aiding in action potential repolarization and maintaining the resting state. Na_v_1.6 sodium channels aggregate at the axon initial segment of cortical neurons, contributing to spike initiation and repetitive discharge properties. Deficiency in Na_v_1.6 subunits results in a hyperpolarized voltage dependence of activation, affecting the spike threshold [18]. Furthermore, persistent sodium currents (I_NaP_) are crucial for neuronal firing, and their dysregulation can lead to abnormal activity, as seen in epilepsy. The differential expression of sodium channels in the neuronal compartments implies that genetic changes affecting I_NaP_ can have opposite effects, promoting bursting activity in the cell body and dendrites but reducing network activity in the axons [19,20]. Aβ-induced neuronal hyperexcitation may be partially attributed to an increased I_NaP_, since soluble Aβ_1–42_ has been shown to increase I_NaP_, and the use of riluzole, an I_NaP_ antagonist, can inhibit Aβ-induced abnormal neuronal activity [20].

However, other studies have shown that as Aβ oligomers accumulate in an AD mouse model, synaptic potentiation in neocortical pyramidal neurons is significantly reduced, primarily due to the disruption of postsynaptic AMPA receptors that leads to a constant diminishing plasticity. In a study examining the septo-hippocampal structure of young AD transgenic mice, notable changes in neural activity were observed. The results indicated a significant reduction in the coordination between individual cell activity in the lateral septum and network activity in the hippocampal CA1 region in these young AD mice. Importantly, these disruptions were evident even before the manifestation of cognitive impairments, underscoring the early impact of AD pathology on neural connectivity [21].

In an effort to find the underlying mechanisms of network hyperactivity in AD, Brown et al., in 2011, used transgenic mice that overproduce Aβ peptides (PSAPP) and described alterations in intrinsic excitability. These mice displayed increased action potential burst and spike afterdepolarization, as well as alterations in sodium and potassium currents, which correlated with Aβ accumulation and neurological deficits. Even though there was no difference in the action potential threshold or the membrane potential span between the resting state and spike threshold, there were significant differences in the rate of rise, zenith, and width of action potentials that could explain neuronal hyperactivity [22]. One year later, it was demonstrated that soluble Aβ has the capability to induce the increase in neuronal firing frequency early in the course of the disease [23]. Similar results were received by studies to another transgenic mouse model (PDAPP), in which no significant alterations were recorded in the subthreshold intrinsic properties such as resting potential, input resistance, membrane time constant, and sag. However, as in pyramidal neurons of PSAPP mice, larger fast after-depolarizing potential, initial higher frequency firing, and narrower spikes were observed [24].

Furthermore, the dendrites of the CA1 pyramidal neurons in mice overexpressing Aβ were shown to exhibit hyperexcitability due to depletion of a dendritic potassium channel (K_v_4.2), and the Tau protein was required for this effect. Increased dendritic excitability, along with changes in ion channels, were considered as key elements leading to neuronal hyperexcitability in the early stages of AD [25]. The role of potassium channels was also highlighted in another study, showing that increased excitability of hippocampal pyramidal neurons was caused by Aβ through alterations in potassium channels. Resveratrol, a natural compound that could reverse Aβ-induced neuron hyperexcitability, did so by restoring the function of potassium channels [26]. Dendrites of hippocampal CA1 pyramidal neurons contain a high density of transient A-type potassium channels, which limit the initiation of action potentials in dendrites, restrict the backpropagation of action potentials, and reduce excitatory synaptic events. These channels play a vital role in regulating the propagation of dendritic potentials [27]. Aβ was found to selectively degrade the highly conserved A-type K^+^ channel, leading to pyramidal neurons’ hyperexcitability in combination with a theta band power decrease during the neuronal spiking [28]. Application of Aβ could also cause a spike broadening in the pyramidal neurons of the hippocampus, an effect mediated by calcium-activated potassium channels, which can be reversed by the suppression of these channels [29].

### 2.2. Interneurons

Interneurons are essential cells distributed throughout the central nervous system (CNS), interacting with principal cells and modulating their excitability. The predominant class of interneurons use neurotransmitter gamma-aminobutyric acid (GABA) to cause inhibition to their targets via opening chloride/bicarbonate or potassium ion channels postsynaptically. Furthermore, whole-cell recordings in current-clamp mode revealed that interneurons showed sustained high firing rates with pronounced after-hyperpolarization. Interneurons had a more depolarized resting membrane potential, smaller action potential amplitude, and larger input resistance compared to pyramidal cells (Figure 2) [30]. Gang Li et al. had formerly shown altered GABAergic input on newborn granule cells leading to delayed maturation of these neurons in 2–3-month-old mice with ε4 allele of apolipoprotein E (APOE4) knock-in. However, they observed decreased frequency of GABAergic miniature spontaneous synaptic currents on newborn neurons, suggesting the hypoactivation of GABAergic interneurons in dentate gyrus [31]. One neurophysiological outcome of this targeted inhibition is to synchronize large principal cell populations at theta (4–8 Hz) and gamma (30–100 Hz) frequencies and regulate their ensemble activity [32]. AD is characterized by abnormal neuronal network activity, including changes in specific brain region activation, disruptions in oscillatory rhythmic activity, and network hypersynchrony [33,34]. The expression of APOE4 and the accumulation Aβ contribute to impaired GABAergic transmission, causing an imbalance between excitation and inhibition and neuronal hyperexcitability [35]. Maintaining a balance between excitation and inhibition is essential for network oscillations, and a disruption in inhibitory control can lead to shifts in balance, causing notable modifications in theta and gamma oscillations [14]. Clinically, AD patients exhibit a higher incidence of seizures, and epileptiform activity is associated with increased Aβ production further contributing to disease progression [36]. Mouse AD models with increased Aβ precursor protein (APP) or APOE4 expression and Aβ or Tau accumulation show network hypersynchrony, spontaneous epileptiform activity, and inhibitory dysfunction [14]. Interventions that enhance GABAergic transmission, such as benzodiazepam treatment, restore network dynamics and coherence in transgenic mouse models [23,37].

Parvalbumin- (PV) and somatostatin- (SST) expressing neurons are two major subclasses of GABAergic interneurons that were found to exhibit network alterations in AD [38]. The first class has a fast-spiking pattern and provides somatic inhibition to postsynaptic neurons [39,40], whereas SST-expressing neurons exhibit a delayed electrophysiological pattern and inhibit distal dendritic parts of postsynaptic neurons [41]. Currently, interneuron electrical patterns alterations and possible ways of restoration seem to be in the center of recently discussed AD research concepts [42,43].

PV interneurons are characterized by both their short action potential duration and their ability to fire at high frequencies, converting an excitatory input signal into an inhibitory output [44]. PV neurons are impaired in APP mice and exhibit reduced expression of the Na_v_1.1 sodium channel subunit, with a decrease also observed in the post-mortem brain tissue of AD patients. Overexpressing Na_v_1.1 in APP mice rescued network imbalance, Gamma power, and memory impairment [45]. While some studies have reported a decrease in the number of PV neurons in AD mouse lines [46,47], the correlation between PV expression and GABAergic activity in AD requires further exploration, since other studies have not shown any reduction in the number of PV interneurons [48,49]. However, a consistent finding is a decrease in inhibitory transmission and activity in AD, potentially linked to the specific impairment of PV neurons, contributing to a failure of inhibitory control, oscillatory changes, increased excitation, epileptic activity, and cognitive decline [50]. In the hippocampus of early stage transgenic APP/PS1 mice (3 and 4 months), PV neurons transiently exhibited hyperexcitability, contributing to heightened network inhibition, while pyramidal neurons showed no significant electrophysiological alterations, with increased spontaneous inhibitory postsynaptic currents indicating augmented inhibitory transmission from hyperexcitable PV interneurons at this stage [47,49]. On the other hand, in 6-month-old animals, the activity of PV interneurons was found to be decreased along with hyperexcitability of pyramidal neurons [49]. At a later stage (7 months), PV neurons became hypoactive, while pyramidal neurons showed hyperexcitability [47,49]. These observations led to the conclusion that PV interneurons in the APP/PS1 mice model of AD become hyperexcitable before any changes are observed on pyramidal cells; thus, the pharmacological restoration of PV interneuron activity could prevent memory loss in the animal models [49]. Inhibiting PV neuron activity early or stimulating it later restores Morris water maze performance in APP/PS1 mice, linking both states to memory impairment in AD. The model suggests soluble Aβ initially enhances PV cell excitability, later affecting pyramidal cells [47,49]. Moreover, dysfunction of hippocampal neuronal network following PV interneuron hyperexcitability was found to increase neuronal vulnerability to Aβ, possibly contributing to AD pathogenetic cascade [51]. Reduced PV neuron activity is associated with decreased Na_v_1.1 and K_v_3 channels, consistent with patient and mouse data showing altered gamma oscillations and increased epileptic activity [49,51]. Furthermore, upon comprehensive examination, it was established that the 5XFAD transgenic mouse model exhibited a notable decrease in the quantity of PV-positive (PV+) interneurons across the entire hippocampal formation compared to both the WT and transgenic Tg4-42 mice [52]. Deficits in the sodium channel subunit Na_v_1.1, prominent in PV-expressing interneurons, are linked to altered network activity and cognitive dysfunction in AD transgenic mice [45,53]. Strategies targeting PV interneuron hyperexcitability may offer long-term benefits, impacting memory, hippocampal network activity, and potentially reducing Aβ plaque deposition. Limited knowledge exists regarding the direct molecular interactions of Aβ with receptors or channels on GABAergic neurons, including PV interneurons. Beta-site Amyloid precursor protein Cleaving Enzyme 1 (BACE1) expression correlates with Na_v_1.1, and Aβ influences synaptic inputs to and from PV interneurons. Aβ induces increased excitability in excitatory pyramidal cells within the anterior cingulate cortex (ACC) by specifically diminishing inhibitory synaptic inputs from fast-spiking (FS) interneurons rather than non-FS interneurons. The primary cause of this disruption in inhibitory input is the perturbation of presynaptic GABA release. Similar results can be observed in the somatosensory cortex of transgenic rTg4510 mice at the age of 5–6 months, where slice patch clamp recordings revealed significant decreases in action potential during positive current injections in fast-spiking (FS) neurons, but not in non-FS neurons. Additionally, FS neurons exhibited increased firing thresholds and decreased maximum re-polarizing slope, indicating reduced excitability selectively in this subtype. These findings suggest hypoactivity specifically in FS neurons, highlighting potential alterations in cortical inhibitory circuits in rTg4510 mice (Figure 3) [54]. Furthermore, it has been identified that the excessive activation of dopamine D1 receptors on FS interneurons contributes to the Aβ-induced disruption of inhibitory innervation. It remains uncertain whether similar dynamics occur in other brain regions [55].

Similarly, both cholecystokinin (CCK) and SST interneurons displayed intrinsic membrane hyperexcitability, characterized by reduced firing threshold, increased membrane input resistance, time constant, and action potential firing frequency in APP knock-in mice compared to age-matched wild-type controls. In contrast to the intrinsic hyperactivity observed in CCK and SST cells, the intrinsic membrane properties of calretinin (CR) expressing cells remained unchanged in both younger (1.5–2 months) and older (9–18 months) APP mice [56]. In 5XFAD mice, the CA1 dorsal part had fewer CR interneurons than WT and Tg4-42 mice, with a similar reduction across the entire CA1 area. In the frontal CA2/3 region, 5XFAD mice showed fewer CR cells compared to WT, but no changes were observed in CA2/3. The DG had unchanged CR cell numbers, and overall, 5XFAD mice had significantly fewer CR cells in the hippocampus compared to WT mice [52]. Reduced numbers of CR cells were observed in plaque-bearing 5XFAD mice, with no significant changes in the Tg4-42 mouse model compared to WT [52]. Finally, anxiety-like behavior in 5xFAD mice has been appeared with an excitatory/inhibitory (E/I) imbalance specifically in the ventral hippocampus (vHPC). The number of PV and SST neurons decreased in the vHPC of these mice, while no reductions were noted in CR cells. Anxiety-like behaviors and the E/I balance in 5xFAD mice were normalized by selectively inhibiting vHPC pyramidal cells through hM4Di expression. Additionally, activating ventral hippocampal SST or PV neurons via a selectively expressed human modified muscarinic receptor (hM3Dq) alleviated anxiety-like behaviors and the synaptic E/I imbalance in the vCA1 of 5xFAD mice. These findings indicate that the anxiety-like behaviors and hippocampal synaptic E/I imbalance in 5xFAD mice result from the loss of SST and PV interneurons in the vHPC, offering insights into the heightened anxiety levels observed in early stage AD patients [57].

## 3. Insights on Neuronal Networks Firing Alterations in AD from Studies on Long-Term Potentiation (LTP)

The phenomenon of LTP and its underlying mechanisms have undergone thorough investigations in the context of AD (Figure 4). These studies have predominantly concentrated on the experimental concepts of induced synaptic strengthening, providing only peripheral insights into the spontaneous activity of neurons and networks. In this review, our focus is exclusively on studies that significantly contribute to the understanding of microscale neuronal electrophysiology in AD. The direct correlation with cellular changes in synaptic plasticity mechanisms is now widely acknowledged. For instance, alterations in the number and functioning of synapses in regions linked to learning and memory, like the hippocampus, have been noted. Older animals exhibit a higher threshold for LTP induction compared to younger ones. These findings are supported by studies indicating increased difficulty in sustaining LTP in the DG and CA3 cells of aging rats, suggesting an age-related decline in the late phase of LTP. Additionally, older animals display enhanced LTD induction at CA3-CA1 synapses due to alterations in Ca^2+^ homeostasis. Hence, the observed deficiencies in synaptic plasticity during aging likely stem from AMPAR trafficking dysregulation. LTP studies on APP transgenic mouse models for AD have yielded conflicting results, with some reporting decreased LTP at a certain age [58,59] and others finding unaltered LTP in the same mice [14,60]. Similar discrepancies exist in APP/PS1 mice [59,61,62], with conflicting reports of reduced LTP by 3 months of age. Synaptic dysfunctions in Tau transgenic AD models are not extensively studied, and few definitive conclusions can be drawn [63,64]. The relationship between morphological changes and electrophysiological properties of CA1 pyramidal neurons is of significant interest, along with their connection to cognitive impairment in AD [63].

In vivo and in vitro experimentations have demonstrated that AD results in the persistent stimulation of NMDARs, LTP, and alterations in cognitive function [65,66]. Moreover, the decline in NMDAR levels contributes to reduced Ca^2+^/calmodulin-dependent protein kinase II (CaMKII) levels, subsequently leading to the deactivation of cyclic AMP response element-binding protein (CREB) and ultimately causing a decrease in the synthesis of brain-derived neurotrophic factor (BDNF), a crucial neurotrophin for synaptic function and the regulation of NMDAR levels [67]. Investigating the mechanisms that lead to the downregulation of LTP, brief exposure to a low concentration of Aβ_1–42_ (1 μM) markedly hindered the initiation of LTP in excitatory input, with no impact on basal transmission, paired-pulse facilitation, NMDAR-EPSCs, or voltage-dependent calcium channel (VDCC) currents. This suggests that the impediment of LTP induction by Aβ is not linked to NMDARs or VDCCs [68].

AMPA receptors (AMPARs) are vital for learning and memory, mediating excitatory neurotransmission in the CNS. In AD, there is a proposed decline in AMPARs and disruptions in LTP/LTD processes. Aβ treatment in neurons reduces AMPAR surface expression, leading to GluA2-specific endocytosis and an imbalance in AMPAR traffic, disrupting Ca^2+^ balance and altering neuronal excitability. Studies on AMPARs in transgenic models (in vitro or in vivo with Tau, APP, or ApoE) can reveal mechanisms underlying susceptibility to LTD in early AD. In parallel, Group I metabotropic glutamate receptors (mGluRs), particularly mGluR1 and mGluR5, activate phospholipase C (PLC) and regulate synaptic plasticity and glutamatergic excitability. In AD, reduced mGluR1 activity is observed in the cortex, potentially enhancing amyloidogenic APP processing. Conversely, mGluR2 is upregulated in the hippocampus of AD patients, with neurofibrillary tangles coexisting in brain regions with mGluR2 overexpression. Studies have suggested a connection between mGluRs and AD pathogenesis, influencing APP processing via α-secretase [69].

Chang et al., using a double knock-in mouse model of AD, with mutations both for APP and presenilin-1, showed that AMPA receptors’ EPSPs exhibited an age-related decrease in currents in the presence of an NMDA receptor antagonist [70]. Moreover, after the stimulation of Schaffer collaterals axons and AMPA receptors’ EPSPs recordings in CA1 cells, the authors reported reduced evoked EPSPs, spontaneous miniature EPSPs, and evoked field EPSPs [70]. The Tau protein seems to affect LTP in hippocampal slices from 4 to 6 months of knock-out for Tau protein mice that were exposed in Aβ_1–42_. After stimulation of Schaffer collaterals, the field EPSPs were recorded in the stratum radiatum of CA1. Importantly, impairment of hippocampal LTP was not evident in mice lacking Tau (TAU^−/−^) when exposed to both rodent and human versions of Aβ_1–42_, while LTP was compromised in slices from wild-type mice [71]. In addition to NMDA and AMPA receptors, nAChRs and neurotensin receptor 1 (NTR1) seem to play vital roles. Local nAChR currents in CA1 induced different outcomes depending on the timing relationship with mild electrical stimulation. Coinciding or preceding nAChR-induced action potentials by 1–5 s led to long-term potentiation, while action potentials within 1 s before electrical stimulation resulted in long-term depression. Outside these time frames, a mismatch of nAChR activity and stimulation produced short-term potentiation. This suggests that ongoing nAChR activity modulates glutamate transmission impact, influencing various forms of synaptic plasticity in the hippocampus [72]. In general, NTR1 receptors enhance the membrane excitability of CA1 pyramidal neurons in hippocampal slices by reducing the spike threshold, possibly by inhibiting voltage-gated K^+^ currents. Furthermore, NTR1 reversed impairments in LTP induced by Aβ_1–42_ oligomers. The protective effect of NTR1 against Aβ-induced LTP impairment was eliminated when spontaneous firing activity was reduced with 10nM tetrodotoxin (TTX), while oral administration of NTR1 enhanced the learning performance of the APP/PS1 mouse model of AD [73].

Early electrophysiological studies in AD identified Aβ as a detrimental factor for LTP. PDAPP mice, expressing a mutant APP form, showed age-dependent Aβ plaque deposition, impacting synaptic field potentials and suppressing LTP in older mice. Exposure to low Aβ concentrations inhibited LTP induction in hippocampal slices. Notably, Aβ_25-35_ and oligomeric Aβ_1–40_, but not fibrillar Aβ_1–40_, attenuated LTP in rat hippocampal slices, confirming the role of Aβ oligomers in early physiological imbalances [74]. Studies have emphasized the importance of Aβ_1–42_ in LTP impairment and have suggested potential treatments to reverse this effect [75,76,77,78]. Finally, age and the combination of genes which regulate APP have different effects on LTP suppression. In APP23 transgenic mice, age showed no significant impact on LTP, except for a notable reduction at 6 months, particularly evident during the induction phase. Prefrontal cortex stimulation in these mice resulted in heightened field potentials [79]. Conversely, in APP/PS1 mice, cognitive impairment and hippocampal LTP deficits emerged progressively from 6 to 8 months, accompanied by a decrease in NMDAR-mediated spontaneous excitatory postsynaptic currents (sEPSCs) [80].

## 4. Neurophysiological Impact of Morphologic and Synaptic Alterations on Neurons and Neuronal Circuits

Neuron cells have distinct molecular, morphological, connectional, and functional properties, and they are characterized by their ability to modify their activity as a response to intrinsic or extrinsic stimulation by reconstructing their architecture, functions, and connections [81]. This feature is well known as neuronal plasticity and constitutes a fundamental property of the nervous system [82,83]. It has been shown that AD pathology is associated with aberrant neuronal excitability along with alterations in synaptic plasticity [84,85]. This abnormal electrical activity, caused by synaptic decay, constitutes the main pathophysiological characteristic of cognitive dysfunction, and it is related with quantitative and qualitative modifications in dendrites and dendritic spines [84,86]. 

Molecular pathways, particularly Rho and Ras GTPases, regulate dendritic spine morphology bidirectionally during synaptic activity. Synaptic plasticity, such as LTP, promotes spine enlargement, and new spine formation, while LTD results in shrinkage and retraction [87,88,89]. Dendritic spine irregularities can disrupt synaptic connectivity and neural circuitry, impacting cognitive and motor skills. Postmortem brain samples from patients with AD exhibit synapse and dendritic spine alterations in the hippocampus and cortex, related to the severity of cognitive impairment [87,88,89]. That is highlighted by several examples: (i) Aβ oligomers induce dendritic spine loss and aberrant dendritic spine formation in primary hippocampal neurons [90,91]; (ii) Tg2576 mice exhibit reduced spine density in CA1 and dentate gyrus, correlating with cognitive decline [92,93]; (iii) the overexpression of human APP in AD animal models leads to substantial spine loss and neurite dystrophy around amyloid plaques, disrupting neuronal circuits and causing cognitive decline [94,95,96]. Elevated soluble Aβ levels have been shown to induce synaptic depression, removal of AMPA receptors, and subsequent synapse and spine loss. Aβ toxicity in dendritic spines involves signaling pathways to Tau and PSD-95, with Tau kinases like PAR-1/MARK and its activating kinase LKB1 implicated in these molecular mechanisms [97]. Cultured neuronal cells from Tg2576 mutant APP transgenic mice exhibit fewer and smaller postsynaptic compartments along with a decrease in PSD-95, highlighting significant Aβ-induced synaptic changes [98].

Consequently, the combination of morphological features and assessment of electrophysiological properties in neurons provides comprehensive information about the progression of AD pathophysiology. Evaluating the effects of Aβ monomers and oligomers on the hippocampal slices of Sprague Dawley (SD) rats, Shankar et al. observed a significant, but reversible, progressive reduction in dendritic spine density in slices treated with Aβ oligomers for 5, 10, and 15 d, whereas Aβ monomers showed no difference [99]. Electrophysiological recordings in hippocampal pyramidal neurons with Aβ oligomers, using whole-cell voltage clamp, showed unaltered membrane capacity but decreased resting membrane resistance and miniature EPSPs amplitude [99]. Furthermore, it was shown that extracellular Aβ oligomers impaired neuronal excitability, reducing AMPA/kainate spontaneous EPSP frequencies and both amplitudes and frequencies of NMDA-evoked EPSPs. Intracellular Aβ oligomers specifically reduced NMDA-evoked EPSP amplitudes. Secreted Aβ disrupted synaptic function, decreasing dendritic spine density in adjacent healthy neurons. Aβ oligomers induced a reduction in LTP in CA1 hippocampal slices from wild-type mice, reversible with β-secretase inhibition, highlighting their role in the amyloidogenic pathway [100].

CA1 pyramidal neurons from P301L pR5 mutant mice (19–20 months of age) appear to have a significant reduction in total dendritic length, especially in distal apical dendrites and in the number of distal apical dendritic nodes. P301L pR5 mice exhibit a notable increase in stubby spines and filopodia across dendritic compartments, while showing no statistical difference in total spine count compared to wild-type. Neurophysiologically, there is a significant reduction in field EPSPs amplitude post-HFS, along with a lowered action potential firing threshold in pR5 neurons of 19–24-month-old mice. Additionally, Tau-mutant pyramidal neurons display an increased current of inward rectifying potassium channels [101]. Young mice (4–5 months old) overexpressing mutant PS1 exhibited increased NMDA receptor-mediated transmission and elevated dendritic spine density per unit length of dendrite in CA1 pyramidal neurons. However, these effects diminished in older (8–10- and 13–14-old-month) mutant PS1 mice and age-matched wild-type mice overexpressing human PS1. In middle-aged (8-10-old-month) mice, both human PS1 and mutant PS1 overexpression correlated with a decline in the number of ramified spines, leading to age-dependent alterations in electrophysiological properties and abnormal synaptic plasticity in younger animals [102]. Establishing the timing of changes in dendritic spine morphology in relation to synapse and neuron loss in APPxPS1 KI mice is important. Research revealed an overall decrease in neuronal density with age; however, this effect was statistically significant in APPxPS1-KI mice, but not in PS1-KI controls. Investigating whether changes in the stratum radiatum occur before neuronal loss, the width of the stratum radiatum was assessed. Similar to observations on neuronal densities, an age-related atrophy of the stratum radiatum was evident in APPxPS1-KI mice, but not in PS1-KI mice. No genotype differences were observed at 3 and 4 months, while at 6 months, APPxPS1-KI mice exhibited a reduction in the width of the stratum radiatum. Evaluation of the morphology of dendritic spines at 3 months by measuring the neck diameter of spines in the plane of electron microscopy (EM) sections demonstrated that APPxPS1-KI mice exhibited an enlargement in the neck diameter of spines in comparison to PS1-KI mice. Consequently, changes in spine morphology preceded the decline in synapse and neuron density. The shortened length and widened width of the spine neck are anticipated to decrease its electrical resistance. APPxPS1-KI mice exhibited a significant reduction in neck resistance. Reducing the spine neck resistance on half had a minimal impact on the simulated EPSP amplitudes measured at the cell body level, regardless of the initial resistance. However, this same reduction notably influenced EPSP in the spine head for neck resistances initially exceeding 80 MΩ. The altered length and increased diameter of necks in APPxPS1-KI mice are also expected to modify the diffusion of molecules across the spine neck. For spines lacking constriction, the head volume was estimated using the spine diameter. Notably, there was a substantial reduction in the time constant in APPxPS1-KI mice, indicating a steep decline in the biochemical compartmentalization of spines. Biopsies from the middle frontal gyrus, particularly in layers II–III, were analyzed to explore potential changes in dendritic spine morphology in individuals with AD. The study involved quantifying stubby spine density and measuring neck diameter for spines with both their neck and postsynaptic density (PSD) in the section plane. Notably, a statistically significant inverse correlation was observed between synapse density and the proportion of stubby spines, as well as between synapse density and neck diameter [103].

In pyramidal neurons of the CA1 region, 4–12-month-old APOE4 mice model of AD showed evidence of significant reduction in dendritic length and dendritic spine head volume and density, compared to APOE3 control neurons. These morphological changes were accompanied by an increase in the slope of field EPSPs and of the ratio P2/P1 in different intervals following repetitive stimulation and paired-pulse facilitation, respectively, in young APOE4 pyramidal neurons. The impaired LTP in neurons from young APOE4 suggests an abnormal synaptic function even in the early stages of AD pathology [104]. In 12-month-old Tg2576 mice with mutant APP overexpression, frontal cortical slices revealed increased dendritic length and volume but decreased spine density in layer 3 pyramidal neurons compared to wild-type mice. Notably, electrophysiological properties, including membrane characteristics and firing frequencies, remained comparable between transgenic and wild-type neurons. Despite reduced dendritic spine density, the maintenance of electrophysiological parameters in Tg2576 neurons may be influenced by increased dendritic volume and relatively low Aβ pathology in this model [105,106].

Another study carried out with a patch clamp on hippocampal slices revealed increased excitability in CA1 hippocampal pyramidal neurons of APP/PS1 mice (4–10 months of age), characterized by increased action potential frequency and burst firing. Morphologically, these neurons exhibited reduced dendritic length and surface area, fewer dendritic branches, and decreased spine density in the stratum lacunosum-moleculare. However, spine density remained unchanged in the stratum radiatum and stratum oriens compared to wild-type littermates [85]. Aβ presence correlates with synaptic depression and dendritic spine loss. In vitro experiments using CA1 pyramidal neuron organotypic slice cultures showed a 31% reduction in dendritic spine density when transfected with APP cDNA and EGFP or incubated with synthetic Aβ_1–42_ for 7 days. Glutamatergic synaptic transmission revealed depression, marked by a reduction in AMPA receptors, emphasizing their role in neuronal morphology and synapses [107].

## 5. Discussion

The typical pattern of AD electrophysiological alterations at neuronal level, according to in vitro and in vivo studies, is characterized by early neuronal hyperexcitability followed by progressive cessation of any electrical activity. Transgenic AD mouse models reveal a link between elevated Aβ levels, aberrant neuronal activity, and cognitive deficits. Hyperactivity of the hippocampal pyramidal neurons in the early stages of the disease is principally driven by soluble Aβ. CA1 neurons exhibit enhanced excitability linked to phosphorylation of Na_v_1.6 and K_v_7 channels. However, the Tau protein is also necessary for these alterations to take place [17]. Dysregulation in sodium currents, particularly the persistent sodium current (I_NaP_), influenced by Aβ, may lead to altered neuronal firing [18,19,20]. Tau plays a crucial role in both decrease in K_v_4.2 levels and associated dendritic hyperexcitability. K_v_4.2 depletion in hAPP/Aβ-expressing mice worsens deficits, particularly in distal dendrites, impacting action potential back propagation crucial for spike timing-dependent plasticity [108,109]. Reduced K_v_4.2 levels contribute to cognitive deficits in hAPP/J20 mice, despite unaffected resting potential and input resistance [109]. Aβ accumulation in transgenic mice leads to increased action potential burst and afterdepolarization. Hippocampal CA1 pyramidal neurons’ dendrites also contain transient A-type potassium ion channels crucial for regulating dendritic potentials [28].

It is also clear that, apart from projection neurons, the interneurons are vulnerable to this process of hyper- and hypo-activation. Further exploration is necessary to understand PV expression and GABAergic activity in AD, as some studies have reported no reduction in the number of PV interneurons [48,49]. Nevertheless, a consistent observation in AD is a decline in inhibitory transmission and activity, potentially linked to the specific impairment of PV neurons. This impairment contributes to a breakdown in inhibition, oscillatory changes, increased excitation, epileptic activity, and cognitive impairment. In the hippocampus of early stage APP/PS1 mice (3 and 4 months), PV neurons briefly demonstrate hyperexcitability, causing a general network inhibition, while pyramidal neurons exhibit no significant electrophysiological changes [47,49]. Conversely, by 6 months of age, PV interneuron activity decreases, accompanied by pyramidal neuron hyperexcitability. At a later stage (7 months), PV neurons become hypoactive, while pyramidal neurons display hyperexcitability. The diminished activity of PV neurons is linked to a reduction in Na_v_1.1 and K_v_3 channels [47,49], aligning with patient and mouse data indicating altered gamma oscillations and increased epileptic activity [49,51]. Modulating PV interneuron activity prevents memory loss, and soluble Aβ enhances PV excitability initially, affecting pyramidal cells [49].

Similarly, both CCK and SST interneurons demonstrated intrinsic membrane hyperexcitability in APP knock-in mice, characterized by a reduced firing threshold, increased membrane input resistance, time constant, and action potential firing frequency. In contrast to the intrinsic hyperactivity observed in CCK and SST cells, the intrinsic membrane properties of CR+ cells remained unaltered in both younger (1.5–2 months) and older (9–18 months) APP mice [56]. In 5XFAD mice, the CA1 dorsal region exhibited a decreased number of CR+ interneurons compared with this reduction to be a consistent evident throughout the entire CA1 area; however, these changes are not visible in other rodent AD models. In the frontal CA2/3 region, 5XFAD mice displayed fewer CR+ cells. The DG showed unchanged CR+ cell numbers, resulting in an overall reduction in CR+ cells in the hippocampus of 5XFAD mice. Notably, plaque-bearing 5XFAD mice exhibited reduced numbers of CR+ cells, while no significant changes were observed in the Tg4-42 mouse model compared to the WT [52]. Limited knowledge exists about direct Aβ interactions with PV interneurons; CCK and SST interneurons display hyperexcitability, while CR cells remain unchanged [56]. The 5XFAD model shows a reduction in PV+ and CR+ interneurons, implicating them in AD-related seizures [52]. It seems this crucial role impairs the sodium channel subunit Na_v_1.1; this is particularly prevalent in interneurons expressing parvalbumin, which are associated with changes in network functioning and cognitive decline in transgenic mice modeling AD [45,53].

It is worth noting that dopamine plays a crucial role in regulating the recurrent excitatory transmission among pyramidal cells and modulating inhibitory inputs from specific interneurons in the brain [110,111]. In AD, Aβ can stimulate the release of dopamine in the frontal cortex by activating α7-nACh receptors [112]. Additionally, the D1 receptor is implicated in Aβ-induced epileptic activity in mice [113], whereas their excessive activation contributes to the imbalance between excitatory and inhibitory activity in the ACC in AD and heightens the excitability of pyramidal cells in the ACC by disrupting inhibitory input from fast-spiking (FS) interneurons through the D1 receptor pathway [55]. Advanced pathology stages showed alterations in serotonin 2A receptor (5HT2AR) density in dopamine and cortical areas. Dopaminergic system dysfunction is observed, with early neuropsychiatric symptoms possibly serving as diagnostic markers. While D2/3R density alterations have not consistently appeared in studies, differences in active/inactive forms of these receptors may exist. In TgF344-AD rats, altered distribution between active and inactive forms of D2R could explain hypersensitivity [114,115,116]. Finally, the TgF344-AD rat model offers insights into AD related neurochemical alterations, particularly in the mesolimbic pathway. Identifying pre-symptomatic D2R-related functional alterations could serve as a clinical diagnostic marker, warranting further investigation in patients [117].

Εlectrophysiological studies focusing on LTP alterations in AD models have offered a wealth of information regarding the impaired synaptic function and reduced neuronal plasticity, even at the early stages of the disease. Although both early and late synaptic responses seem to become depressed, aged tissue derived from wild-type mice exhibited significantly greater depression compared to the transgenic APdE9 perforant pathway circuit (Figure 5) [118]. However, little do these elements add in our understanding on the spontaneous neuronal excitability throughout the progress of AD. In transgenic mouse models like PDAPP mice, age-related Aβ deposition in amyloid plaques suppresses LTP in the hippocampus [72], despite minimal changes in plasticity due to Aβ plaques and neurofibrillary tangles. Furthermore, even Aβ variants lacking neurotoxic effects can hinder LTP, suggesting an impact on synaptic plasticity independent of neurotoxicity [119,120]. The absence of Tau protein appears to influence LTP in hippocampal slices from young Tau knockout mice exposed to Aβ_1–42_. Significantly, the impairment of hippocampal LTP was not observed in TAU^−/−^ mice when exposed to both rodent and human versions of Aβ_1–42_, in contrast to compromised LTP observed in slices from wild-type mice [71]. In a double knock-in mouse model of AD, featuring mutations in both APP and presenilin-1, an age-related reduction in currents of AMPA receptors’ EPSPs in the presence of an NMDA receptor antagonist was observed, with further reductions in evoked EPSPs, spontaneous miniature EPSPs and evoked field EPSPs, recorded after Schaffer collaterals axon stimulation and AMPA receptor EPSPs recordings in CA1 cells [70]. nAChRs and NTR1 receptors play a crucial role in synaptic plasticity beyond that of NMDA and AMPA receptors. Temporally aligned or preceding mild electrical stimulation induces long-term potentiation with local nAChR currents in CA1, while action potentials occurring within 1 s before stimulation lead to long-term depression. This time-dependent modulation underscores the influence of nAChR activity on glutamate transmission and its impact on synaptic plasticity in the hippocampus [72].

Further investigating the cause of neurophysiological alterations in AD, we should highlight that the neuronal morphology and asymmetric development of the dendritic field depend mainly upon compartmentalization of protein localization. During the early stages of AD, there is a notable reduction in hippocampal synapses and synaptic proteins, accompanied by neuronal hyperexcitability [121,122]. CA1 pyramidal neurons in P301L pR5 mice exhibited reduced total dendritic length and altered spine morphology [94,95,96]. Overexpression of APP and PS1 genes in AD rodent models resulted in reduced dendritic spine density [103]. Aβ toxicity induced synaptic depression and dendritic spine loss through the Tau and PSD-95 signaling pathways [97,98]. Aβ oligomers contributed to a reversible reduction in dendritic spine density, possibly dependent on Tau presence [19,20]. Studies on LTP in AD transgenic models have shown conflicting results, with APOE4 mice exhibiting impaired LTP, while Tg2576 mice displayed altered dendritic morphology [104]. In vivo patch clamp studies on APP/PS1 mice revealed heightened excitability and morphological alterations in CA1 pyramidal neurons. Despite reduced dendritic spine density in Tg2576 mice, electrophysiological parameters were maintained, potentially due to the increased dendritic volume and low Aβ pathology [85].

All the data presented in this review paper provide abundant information about the pivotal role of altered neuronal excitability in the initiation and progression of the AD pathogenetic cascade. Therefore, it comes as no surprise that, during the last decade, many research efforts have been focused on methods aiming to modify neuronal activity using non-invasive brain stimulation methods [123]. The induction of high-frequency oscillations has been shown to synchronize hippocampal circuits and may be considered a potential treatment method for AD [5]. Those rhythmic fluctuations of the brain are generated by circuits of fast-spiking neurons, and their impairment may lead to memory dysfunction [124]. The potential of intervening on AD pathology through the manipulation of brain activity is supported by observations on AD models that impaired γ oscillations proceeding from the Aβ plaque formation, whereas by the restoration of those oscillations, Aβ plaques can be significantly reduced [7,123]. Lastly, it is asserted that restoration of the hippocampal activity may also happen through the GABAergic interneuron restoration, while microglia activation can reduce the Aβ plaque burden [125].

## 6. Conclusions

AD manifests a complex pattern of neuronal alterations, beginning with early hyperexcitability, followed by progressive cessation of electrical activity. Pyramidal neurons in the hippocampus undergo changes in membrane properties and spiking patterns, while interneurons are also vulnerable to aberrant activation, which may occur before affecting projection neurons. The disease is characterized by synaptic loss, dendritic spine reduction, and altered glutamate receptor-dependent pathways, unveiling the intricate interplay between Aβ, Tau, and synaptic dysfunction in the pathogenesis of AD. It is evident that each stage of the disease demonstrates specific patterns of electrophysiological alterations, and any intervention with brain stimulation must take those into account for the optimization of efficacy.

## Figures and Tables

**Figure 1 cells-13-00434-f001:**
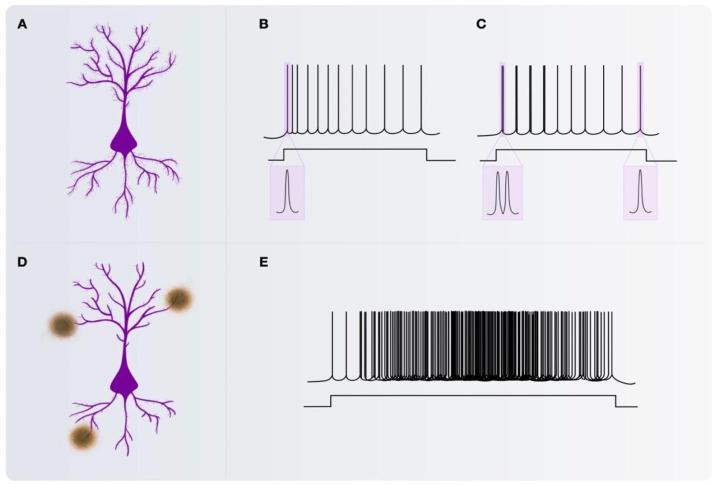
Illustration of pyramidal neurons and their electrophysiological activity pattern. (**A**) Schematic representation of a pyramidal neuron in the CA1 area of the hippocampus. (**B**) Electrophysiological pattern of regular spiking pyramidal neuron. (**C**) Electrophysiological pattern of bursting pyramidal neuron. (**D**) Schematic representation of a pyramidal neuron with reduced dendritic spines in AD. (**E**) Electrophysiological model of a pyramidal neuron in AD.

**Figure 2 cells-13-00434-f002:**
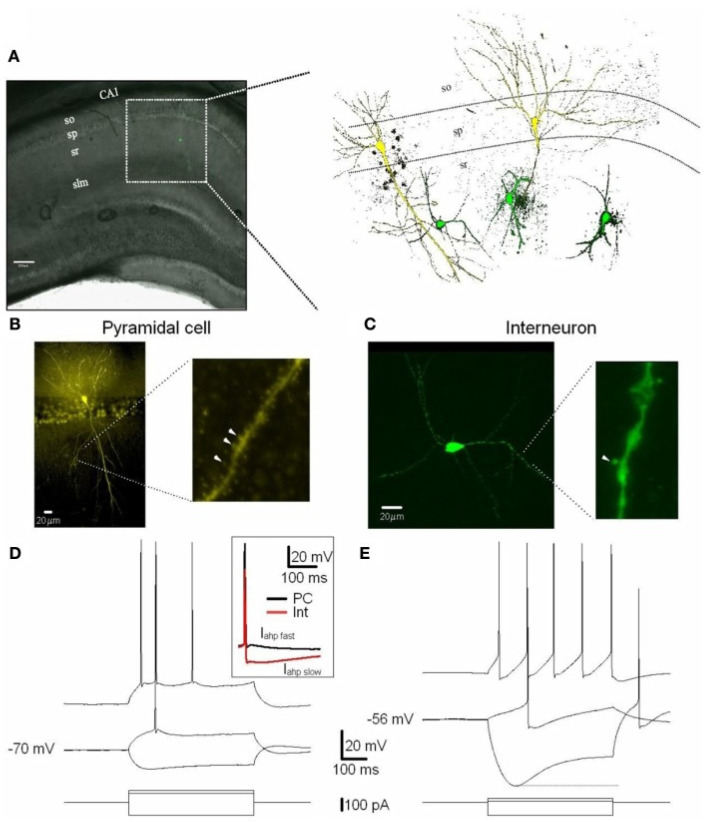
Excitatory (PCs) and inhibitory (Ints) neurons recorded in the CA1 region of the hippocampus. (**A**) CA1 PCs and Ints were patch-clamped with a pipette solution containing Lucifer Yellow (2 mM). The location of neurons in the slices was visualized by superimposition of the reflected light image of the hippocampal slice and of the Lucifer Yellow fluorescence signal (left panel). The right panel shows reconstructed confocal images of two PCs (yellow) and three Ints (green) recorded in the CA1 sp and sr, respectively. (**B**,**C**) Confocal images of one PC (**B**) and one Int (**C**) of CA1 pyramidal layer and stratum radiatum, respectively. (**B**) Right and (**C**) right expanded confocal images of dendritic spines. Note the spiny dendritic segment of the PC (**B**) in contrast with the a-spiny (**C**) one of the Int. The arrowheads indicate dendritic spines. (**D**,**E**) Voltage responses of one PC and one Int (top), to a series of intracellular current pulses (bottom) are shown. The current was applied at rest (−70 and −56 mV for PC and Int, respectively). Inset, action potentials from a PC and Int are superimposed. Note the larger Iahp in Int compared to PC. Abbreviations: so, stratum oriens; sp, stratum pyramidale; sr, stratum radiatum; slm, stratus lacunosum moleculare. Scale bar: 200 μm. (Reproduced with permission from Martina, M., Comas, T., & Mealing, G.A. (2013). Selective Pharmacological Modulation of Pyramidal Neurons and Interneurons in the CA1 Region of the Rat Hippocampus. *Frontiers in pharmacology*, *4*, 24. https://doi.org/10.3389/fphar.2013.00024 [30]. This content is licensed under the Creative Commons Attribution License).

**Figure 3 cells-13-00434-f003:**
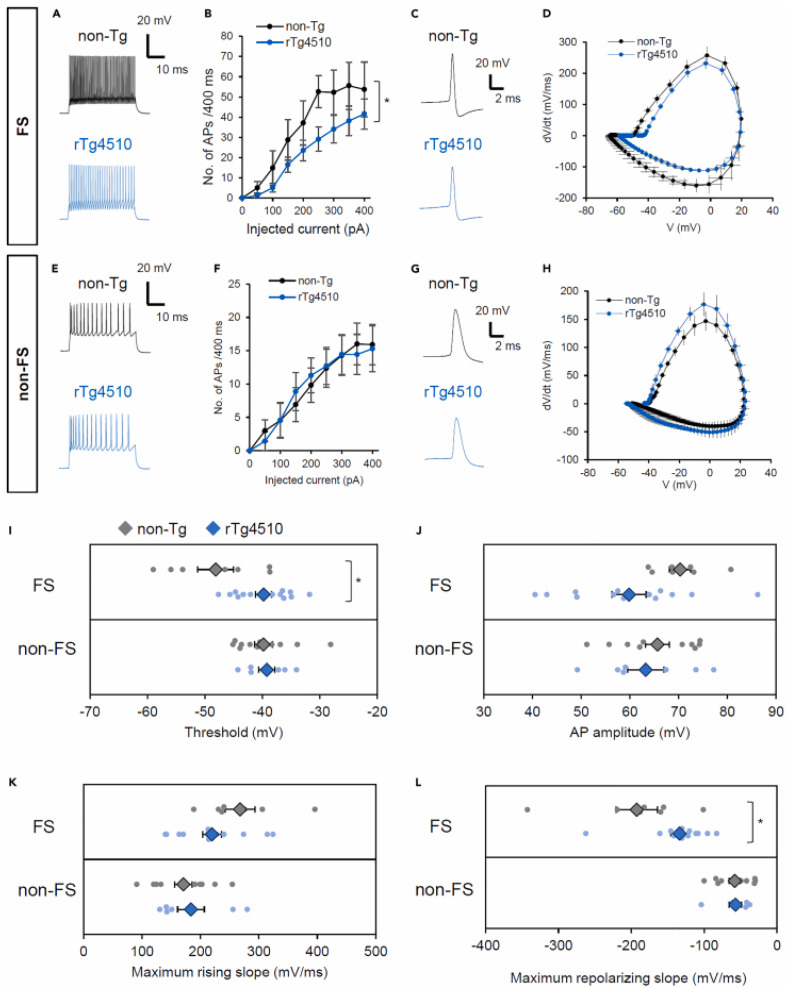
Firing properties of inhibitory interneurons in 5–6-month-old non-Tg and rTg4510 mice. (**A**,**B**) Input–output (I–O) relationship in fast-spiking (FS) neurons. Representative traces evoked by +300 pA injection are shown in (**A**). (**C**,**D**) Single action potentials (APs) and phase plots of FS neurons. A representative single AP is shown in (**C**). (**E**,**F**) I–O relationship in non-FS neurons. Representative traces evoked by +300 pA injection are shown in (**F**). (**G**,**H**) Single APs and phase plots of non-FS neurons. A representative single AP is shown in (**H**). (**I**–**L**) Parameters of single APs: threshold (**I**), AP amplitude (**J**), maximum rising slope (**K**), and maximum repolarizing slope (**L**). Data are shown as mean ± S.E.M. FS, N = 7 cells from seven slices (7 non-Tg mice), N = 13 cells from 12 slices (eight rTg4510 mice); non-FS, N = 11 cells from 6 slices (7 non-Tg mice), N = 7 cells from 7 slices (6 rTg4510 mice). *p* value (* *p* < 0.05) by two-way ANOVA (**B**), or by the Student’s *t* test (**I**,**L**) (Reproduced from Kudo, T., Takuwa, H., Takahashi, M., Urushihata, T., Shimojo, M., Sampei, K., Yamanaka, M., Tomita, Y., Sahara, N., Suhara, T., & Higuchi, M. (2023). Selective dysfunction of fast-spiking inhibitory interneurons and disruption of perineuronal nets in a tauopathy mouse model. *iScience*, 26(4), 106342. https://doi.org/10.1016/j.isci.2023.106342 [54] with permission from Elsevier).

**Figure 4 cells-13-00434-f004:**
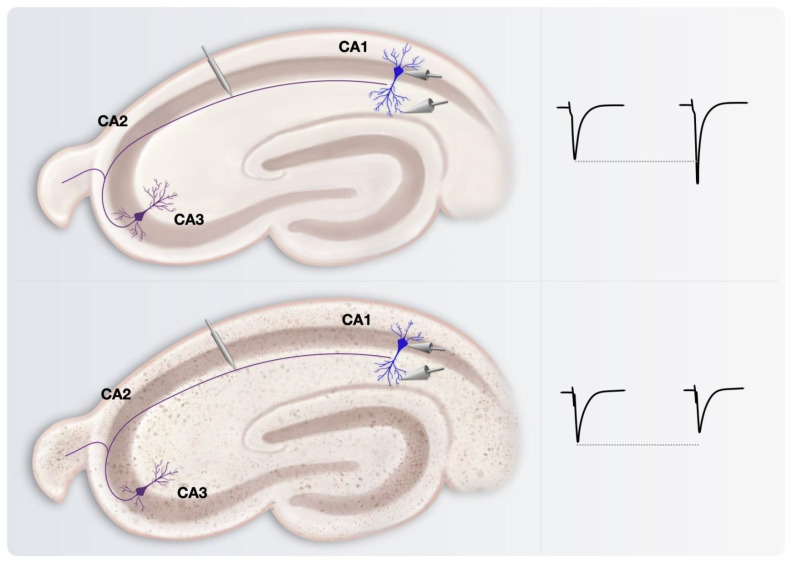
Extracellular recording of the response of CA1 pyramidal neurons to stimulation of afferent Schaffer collaterals. The stimulation evokes an afferent volley) and a field EPSP. After a tetanus (two trains of 100 Hz, 1 s duration, 30 s interval) applied through the same stimulating electrode, the field EPSP is recorded. Note the increased initial slop after the tetanus.

**Figure 5 cells-13-00434-f005:**
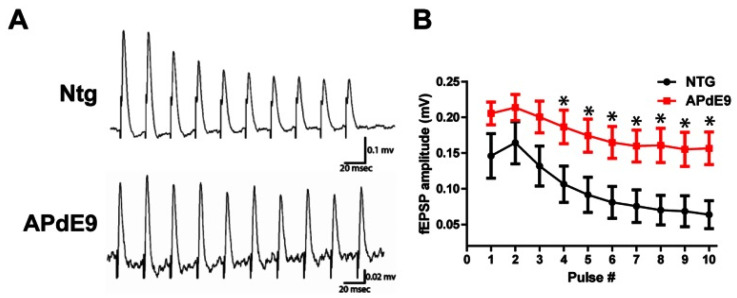
(**A**) Representative field potential responses recorded extracellularly in NTG mouse tissue. A 40 Hz, 10-pulse stimulation evoked depressive (decreasing amplitude) responses in the NTG animals. APdE9 mice showed sustained field potential responses with little decrease in the amplitude throughout the stimulation train. (**B**) Average and S.E.M. for APdE9and NTG fEPSP amplitudes during the 40 Hz stimulation. Note significant differences for the fEPSP amplitudes produced by pulse stimulations 4–10 (marked with *) (pulse to pulse comparison, unpaired *t*-test, *p* < 0.05). (Reproduced with permission from Hazra, A., Gu, F., Aulakh, A., Berridge, C., Eriksen, J. L., & Ziburkus, J. (2013). Inhibitory neuron and hippocampal circuit dysfunction in an aged mouse model of Alzheimer’s disease. *PLoS ONE*, 8(5), e64318. https://doi.org/10.1371/journal.pone.0064318 [118]. This content is licensed under the Creative Commons Attribution License).

## Data Availability

No new data were created or analyzed in this study. Data sharing is not applicable to this article.

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
