# Peer review of "Firing Alterations of Neurons in Alzheimer’s Disease: Are They Merely a Consequence of Pathogenesis or a Pivotal Component of Disease Progression?"

_cells, 2024, doi:10.3390/cells13050434_

Round 1
Reviewer 1 Report
Comments and Suggestions for Authors
Paper is well written and depict clearly the most recent advances in Alzheimer's Disease research firing alteration. Most important topic are addressed. I would just suggest a more detailed discussion about the role of domaine in regulating the neural circuit and the potential role of this neurotransmitter as early marker. Figure 2 is not clear, bigger and/or more detailed traces could be useful.
Author Response
We would like to thank the reviewer for giving us the opportunity to revise the manuscript. We believe that the comments helped us significantly enhance the paper.
Response to reviewer’s comments
Reviewer 1
- I would just suggest a more detailed discussion about the role of dopamine in regulating the neural circuit and the potential role of this neurotransmitter as early marker.
We have added a detailed discussion about the role of dopamine. (lines 715-730)
- Figure 2 is not clear, bigger and/or more detailed traces could be useful.
We are uploading a high-resolution image for Figure 2 along with the revised manuscript.
Reviewer 2 Report
Comments and Suggestions for Authors
Dear the authors,
Tzavellas et al. reviewed in the current study fire alterations of Neurons in AD. Basically, this review manuscript is very interesting. However, the authors should address some questions. The comments are listed as below.
1. The reviewer guess the current manuscript is a kind of review article, but why do the authors separate the RESULT and DISCUSSION sections? I do not know what is the RESULT. Did the authors do any experiments? I also do not understand what the authors contributions on line626 are. Please check again about the content.
2. Overall, the paper has few figures. A few more figures would be more effective.
3. The RESULTS and DISCUSSION are redundant. It would be better to organize them a little more.
4. The authors should have a proofread in English. Most of them are fine, but there are a few errors.
Comments on the Quality of English LanguageThe authors should have a proofread in English. Most of them are fine, but there are a few errors.
Author Response
We would like to thank the reviewer for giving us the opportunity to revise the manuscript. We believe that the comments helped us significantly enhance the paper.
Response to reviewer’s comments
Reviewer 2
- The reviewer guess the current manuscript is a kind of review article, but why do the authors separate the RESULT and DISCUSSION sections? I do not know what is the RESULT. Did the authors do any experiments? I also do not understand what the authors contributions on line626 are. Please check again about the content.
We have reorganized the headings of the whole manuscript according to the comments of the reviewer.
- Overall, the paper has few figures. A few more figures would be more effective.
We have added three more figures in the manuscript to address this comment.
- The RESULTS and DISCUSSION are redundant. It would be better to organize them a little more.
We have reorganized the headings of the whole manuscript according to the comments of the reviewer.
- The authors should have a proofread in English. Most of them are fine, but there are a few errors.
We have conducted language editing on the text, implementing grammar and syntax fixes.
Reviewer 3 Report
Comments and Suggestions for Authors
The authors of the manuscript titled 'Firing Alterations of Neurons in Alzheimer’s Disease: Are They Merely a Consequence of Pathogenesis or a Pivotal Component of Disease Progression?' have reviewed specific patterns of electrophysiological alterations in the each stage of the disease Although this subject is of interest to this reviewer, I have several concerns with their figures. In my opinion, this manuscript is not recommended for publication in its present form, but may accept as the paper after major revision.
Comment #1: More figures
Authors should show and make more appropriate figures that support their argument.
Comment #2:
Why does Aβ selectively inhibit or block certain channels? I hope that the authors would be modified this point in the results or discussion section of the revised manuscript.
Minor points
1 Abbreviations
Abbreviations used should be defined once the first time they appear in the text.
Ex l 112; β-amyloid, l122; β-amyloid, l165; PDAPP, l190; CNS l210; APP, l233; APP/PS, l252; 5xFAD, l254; Tg4-42, l260; BACE1, l394 NTR1, and so on
2 L280 CR- cells ~ (n=10). (n=8). What does this mean CR-? nagtive?
3 L212 dysfunctio 41→dysfunction 41
4 Please review the grammar and spelling again.
Comments on the Quality of English LanguagePlease review the grammar and spelling again.
Author Response
We would like to thank the reviewer for giving us the opportunity to revise the manuscript. We believe that the comments helped us significantly enhance the paper.
Response to reviewer’s comments
Reviewer 3
- Authors should show and make more appropriate figures that support their argument.
We have added three more figures in the manuscript to address this comment.
- Why does Aβ selectively inhibit or block certain channels? I hope that the authors would be modified this point in the results or discussion section of the revised manuscript.
We have added in line 212 the sentence “Αβ was found to selectively degrade the highly conserved A-type K+ channel leading to pyramidal neurons hyperexcitability in combination with theta band power decrease during the neuronal spiking.” to address this comment. This was a selective effect of Aβ on specific channels that we found in the relevant bibliography.
- Abbreviations used should be defined once the first time they appear in the text.
According to the reviewer’s comment we defined all abbreviations the first time they appeared in the text. Unfortunately, for the abbreviated names of the mouse and rat AD models we could not find their full names, as they are widely used in the bibliography without any definition.
- L280 CR- cells ~ (n=10). (n=8). What does this mean CR-? nagtive?
We have updated the names of the interneurons according to the reviewer’s comment.
- L212 dysfunctio → dysfunction
The word has been corrected.
- Please review the grammar and spelling again.
We have conducted language editing on the text, implementing grammar and syntax fixes.
Round 2
Reviewer 3 Report
Comments and Suggestions for Authors
I hvae no consens.